materials science

corn–wheat starch, zein, bilayer films, storage conditions, physico-chemical properties

**Author for correspondence:**
Fusheng Chen
e-mail: fushengc@haut.edu.cn

This article has been edited by the Royal Society of Chemistry, including the commissioning, peer review process and editorial aspects up to the point of acceptance.

# Effect of storage condition on the physico-chemical properties of corn–wheat starch/zein edible bilayer films

Chen Chen, Yan Du, Guanjie Zuo, Fusheng Chen, Kunlun Liu and Lifen Zhang

College of Food Science and Technology, Henan University of Technology, Zhengzhou, Henan Province 450001, People's Republic of China

CC, 0000-0001-5914-228X; YD, 0000-0003-4123-8720; FC, 0000-0002-8201-1234

The functional properties of biopolymer-based film packaging materials are susceptible to external storage conditions. The effects of different storage temperature, relative humidity (RH) and duration on the apparent form, barrier properties, mechanical properties and microstructure of corn–wheat starch/zein bilayer films were studied. From 0 to 150 days, storage temperature and RH, but not storage time, affected the appearance and colour of the bilayer films. The increase in haze of the bilayer films stored at 25°C was much greater than that at low temperatures. With increased storage time, the moisture content first increased and then decreased, while the water resistance and oxygen barrier properties of the bilayer films worsened. After 150 days, the bilayer film stored at 25°C with 54% RH had better water resistance properties. The oxygen barrier properties of the bilayer film stored at 25°C with 43% RH were preferable to those of other groups because the peroxide value of vegetable oil packed in the former bilayer film was the lowest. The tensile strength of bilayer films stored at 25°C with RH of 43, 54 and 65% decreased, but was still better than those stored at low temperatures (−17°C, 4°C), which were tough due to their high elongation at break. Scanning electron microscopy results showed tight bonds between the bilayer films, and the network structure inside the films disappeared and reappeared during storage. The cross-sectional compactness changed, and there was no film separation after 150 days.

# 1. Introduction

Food packaging materials play an important role in extending shelf life and ensuring food safety and quality. Synthetic petroleum-based polymers are widely used in the food packaging industry because of their variety, low price and stable nature. However, the non-degradability of the synthetic petroleum-based polymers and issues related to environmental pollution are major problems that cannot be ignored [1]. With the increasing pollution from petroleum-based polymer packaging materials and the depletion of petroleum resources, new degradable and edible environmentally friendly food packaging materials prepared from biopolymer resources, including polysaccharides, proteins and lipids, have gradually become a popular research topic [2].

Starch, a natural low-cost macromolecular polymer with the ability to biodegrade, has been widely used in edible packaging films [3]. Corn starch (C), wheat starch (W), potato starch and pea starch are common film-forming substrates for starch films [4]. However, due to the crystalline region in starch and the strong water absorption of starch molecules, simple starch films have high brittleness and poor water resistance, limiting their use in practical application [5]. Zein (Z), which contains a large number of hydrophobic and sulfur-containing amino acids, is the main protein in maize [6]. In ethanol solution, Z shows an irregular linear structure, and a transparent and smooth film is formed after solvent evaporation. Due to the interactions of the disulfide bond (S–S), covalent bond, hydrogen bond and the electrostatic force and hydrophobic interaction between protein chains, Z-based films usually have excellent film-forming ability, water resistance, toughness, oxygen barrier properties and biocompatibility [7,8]. However, the poor tensile strength and high price of Z-based films limit their application in the packaging industry [9]. Due to the distinct advantages of starch and Z films, a layer of Z can be added to starch films to make full use of these advantages, which complement each material to prepare edible bilayer films with excellent properties and expanded applications in practical production.

In practical applications, the performance stability of films during storage is the key to ensuring the quality of the stored food. How to maintain the good functional properties of film packaging materials during storage is a difficult problem [10]. The additive ingredients added into the film matrix affect the chemical and physical properties of film packaging materials [11,12]. In addition, physical and chemical changes may occur in the films due to differences in storage temperature, relative humidity (RH), duration, oxygen concentration and light intensity [13,14]. Film properties, such as film colour, tensile strength, elongation at break, solubility, water resistance and oxygen barrier properties, may change accordingly [15,16].

Currently, there are few studies on the changes in the functional properties of bilayer films during storage [17,18]. Therefore, to accelerate the popularization and application of these bilayer films, in this experiment, C–W/Z edible bilayer films were stored for 150 days under freezing (−17°C), refrigeration (4°C) and room temperature (25°C; RH: 43, 54, 65%) conditions. The physical and chemical properties of the bilayer films were determined every 30 days. The effects of different storage temperature, RH and time on the surface colour, transmittance, haze, moisture content, water vapour permeability (WVP), oxygen barrier properties, tensile strength and elongation at break of the bilayer films were investigated in detail. In addition, the changes in the film microstructure were observed by scanning electron microscopy (SEM).

# 2. Material and methods

## 2.1. Materials

C (containing 27.5% amylose) was purchased from Dingfeng Develop Co., Ltd (Tianjin, China). W (containing 29.0% amylose) was obtained from Lvyuan Starch Co., Ltd (Shanghai, China). Z (containing 91.1% protein) was provided by Dulai Biotechnology Co., Ltd (Jiangsu, China). D-Sorbitol, sodium alginate, citric acid and carboxymethyl cellulose sodium were provided by Tianjin Chemical Reagent Co., Ltd (Tianjin, China). Ethyl alcohol, polyethylene glycol-400 and glycerol (analytical grade) were purchased from Luoyang Reagent Co., Ltd (Henan, China). All other reagents were provided by Luoyang Reagent Co., Ltd (Henan, China). All of the reagents were of analytical grade or better.

## 2.2. Preparation of C–W film

C and W were used as film-forming substrates (mass ratio 2 : 3; 6.00 g/100 ml), followed by the addition of sorbitol (0.6% w/v; 0.60 g sorbitol in 100 ml distilled water), citric acid (2.5% w/v; 2.50 g citric acid in

100 ml in distilled water) and a mixture of carboxymethyl cellulose sodium and sodium alginate at a 3 : 2 (w/w) ratio (1.4% w/v; 1.40 g mixture in 100 ml distilled water). The mixture was stirred at 40°C for 20 min by a water bath oscillator (SHA-2D, Changzhou Aihua Instrument Manufacturing Co., Ltd, Jiangsu, China). Then, the film-forming solution was heated to 85°C and kept for 40 min for gelatinization. Subsequently, a vacuum pump (DZF, Shanghai Binglin Electronics Technology Co., Ltd, Shanghai, China) was applied to de-aerate the film-forming gel at 80°C and 0.09 Pa for 30 min. After casting 10 ml of the gel onto plastic dishes with diameter of 100 mm, the C–W film layer was first formed. Then, the film was dried at 45°C for 12 h to complete the preparation.

## 2.3. Preparation of Z film-forming solution

The 6.00 g of Z was dissolved in 100 ml of aqueous ethanol (95%, v/v) for the preparation of Z film-forming solution. The mixture was stirred using an electric stirrer for 3 min (FJS-4, Changzhou Dingxin Experimental Instrument Co., Ltd, Jiangsu, China). Then, 0.90 g of polyethylene glycol 400 and 1.80 g of glycerol were added, followed by stirring for 20 min. Finally, a thermostat (XMTD-7000, Forvo Electronic Technology Co., Ltd, Hunan, China) was used to incubate the solution at 80°C for 15 min.

## 2.4. Preparation and storage of C–W/Z bilayer films

A C–W/Z bilayer film was prepared by adding 3 ml Z film-forming solution to the dried C–W film and drying the mixture for 1.5 h with an 80°C blast after 30 min. Then, the bilayer film was stored for 30, 60, 90, 120 and 150 days under freezing (−17°C), refrigeration (4°C) and room temperatures (25°C; RH: 43, 54, 65%).

## 2.5. Measurement of the surface colour of the C–W/Z bilayer films during storage

The surface colour of the C–W/Z bilayer films was measured using the method reported by Pereda et al. with few modifications [19]. The film colour was tested by a WSC-S colour difference meter (Changzhou Nuoji Instrument Co., Ltd, Jiangsu, China). The equation for the calculation of the whiteness (WI) of the film was as follows:

$$WI = 100 - [(100 - L^*)^2 + a^*2 + b^*2]^{0.5}, \tag{2.1}$$

where the $a^*$ value represents red (+) to green (−), the $b^*$ value represents yellow (+) to blue (−) and $L^*$ is 0 for black and 100 for white. Triplicate films were prepared and quintuple measurements were performed for each film.

## 2.6. Measurement of the transmittance and haze of the C–W/Z bilayer films during storage

Bilayer films with smooth surfaces and without contamination were selected, and the transmittance and haze of the C–W/Z bilayer films were measured by a WGT-S transmittance/haze meter (Shanghai Jingke Industrial Co., Ltd, Shanghai, China). Each film sample was measured in three parallels.

## 2.7. Measurement of the moisture content of the C–W/Z bilayer films during storage

The moisture content of the C–W/Z bilayer films was detected gravimetrically. The bilayer films were dried at 105°C in an oven (DHG-9030A, Gongyi Honghua Instrument Equipment Industry and Trade Co., Ltd, Henan, China) for 24 h. All of the detections were performed in triplicate.

## 2.8. Measurement of the WVP of the C–W/Z bilayer films during storage

The WVP was measured gravimetrically using the method reported by Talja et al. with few modifications [20]. A 20.00 g of anhydrous calcium chloride was accurately weighed in an aluminium box with a diameter of 55 mm. Then, the aluminium box was completely sealed with bilayer films and placed in a dryer containing saturated sodium chloride solution (25°C, RH: 75%). The mass of anhydrous calcium chloride was determined accurately until its mass was constant weight. The WVP was calculated with equation (2.2). The water resistance of the bilayer films was evaluated based on the

degree of WVP.

$$\text{WVP} = \frac{mL}{At\Delta p}, \tag{2.2}$$

where $L$ represents the thickness of the bilayer film (µm), $m$ represents the increased mass of anhydrous calcium chloride after water absorption (g), $A$ represents the permeation area (m$^2$), $t$ represents the time of permeation (s) and $\Delta p$ represents the water vapour pressure difference across the bilayer film (2376.30 Pa). Each sample was repeated three times.

## 2.9. Measurement of the oxygen barrier properties of the C−W/Z bilayer films during storage

First, 20.0 g of soya bean oil was packed in a 250 ml of conical bottle, sealed with the bilayer film and aged in a 60°C incubator for 10 days. Then, the peroxide value (POV) of the aged soya bean oil was determined according to Chinese National Standard GB 5009.227–2016. The oxygen barrier properties of the C−W/Z bilayer films were evaluated based on the size of the POV.

## 2.10. Measurement of mechanical properties of the C−W/Z bilayer films during storage

At first, the films were cut into strips (10 × 50 mm) and mounted between a TA 96 tensile grip (Stable Micro Systems, UK). Then, a TAXT-plus texture analyser (SMS, UK) containing a load cell of 1 kg was applied for the determination of the tensile strength and elongation at break of the C−W/Z bilayer films. The cross-head speed was set at 1.0 mm s$^{-1}$ and the initial grip separation was set at 30 mm. All of the detections were performed in triplicate.

## 2.11. SEM of cross-sectional C−W/Z bilayer films during storage

The cross-sectional structure of the C−W/Z bilayer films was tested using a Quanta Feg 250 instrument (FEI Company, USA) at an accelerating voltage of 20 kV. The C−W/Z bilayer film specimens were cut into small pieces (5 × 5 mm). Each piece was sputter coated with gold before testing.

## 2.12. Statistical analysis

Statistical analysis was performed using SPSS Statistics 17.0 (IBM Corp., New York, USA)

Analysis of variance was conducted to determine the significance of colour, transmittance, haze, moisture content, WVP, oxygen barrier properties, tensile strength and elongation at break among the bilayer films. Significant difference ($p < 0.05$) between means was measured using Duncan's multiple range test.

# 3. Results and discussion

## 3.1. The effect of storage conditions on the surface colour of the C−W/Z bilayer films

Analysing changes in the surface colour of films during storage is of great significance for studying the shelf life of convenient food packaging materials [21]. Tables 1–4 show the changes in the $L^*, a^*, b^*$ and WI values of the bilayer films during storage. The $L^*$ values of the bilayer films did not change significantly with prolonged storage time but decreased slightly after 120 days. Moreover, during the same storage period, changing the temperature and RH did not cause a significant change in the $L^*$ values of the bilayer films, indicating that the brightness of the bilayer films did not change significantly during the storage process. The $a^*$ values of bilayer films decreased slightly with prolonged storage time but increased significantly with increasing temperature and RH during the same storage period ($p < 0.05$). Under storage conditions of 25°C with 65% RH, the $a^*$ values reached the maximum values of −1.50 (30 days), −1.66 (60 days), −1.81 (90 days), −1.69 (120 days) and −2.05 (150 days), and the green colour of the bilayer films decreased in intensity. Similar to the trend of the $a^*$ values, the $b^*$ values decreased significantly with increasing temperature and RH during the same storage period, and the yellowness of the bilayer films decreased. The increase in $a^*$ values and decrease in $b^*$ values of the bilayer films may be due to the increases in storage temperature and RH and the loosening of the internal structure of the bilayer films, which allowed a part of the Z film to enter the C−W film.

**Table 1.** Changes in the $L^*$ value of the bilayer films during storage. Different lower-case and upper-case letters in the table indicate significant differences ($p < 0.05$) in the same row and column, respectively.

| storage conditions | 30 days | 60 days | 90 days | 120 days | 150 days |
|---|---|---|---|---|---|
| −17°C | 94.47 ± 0.32[b,A] | 94.18 ± 0.24[ab,A] | 93.77 ± 0.31[a,A] | 94.10 ± 0.54[ab,A] | 94.31 ± 0.10[ab,B] |
| 4°C | 94.62 ± 0.14[b,A] | 94.53 ± 0.40[b,A] | 94.64 ± 0.22[b,B] | 94.46 ± 0.56[b,A] | 93.44 ± 0.54[a,A] |
| 25°C, RH: 43% | 94.67 ± 0.33[a,A] | 94.47 ± 0.07[a,A] | 94.22 ± 0.31[a,AB] | 94.35 ± 0.59[a,A] | 94.17 ± 0.31[a,B] |
| 25°C, RH: 54% | 94.75 ± 0.30[b,A] | 94.33 ± 0.22[b,A] | 94.33 ± 0.28[b,B] | 94.37 ± 0.12[b,A] | 93.75 ± 0.41[a,AB] |
| 25°C, RH: 65% | 94.44 ± 0.16[a,A] | 94.34 ± 0.38[a,A] | 94.29 ± 0.24[a,B] | 94.06 ± 0.27[a,A] | 94.08 ± 0.29[a,AB] |

**Table 2.** Changes in the $a^*$ value of the bilayer films during storage. Different lower-case and upper-case letters in the table indicate significant differences ($p < 0.05$) in the same row and column, respectively.

| storage conditions | 30 days | 60 days | 90 days | 120 days | 150 days |
|---|---|---|---|---|---|
| −17°C | −2.90 ± 0.12[a,A] | −3.17 ± 0.20[a,A] | −3.15 ± 0.04[a,A] | −3.11 ± 0.15[a,A] | −3.19 ± 0.19[a,A] |
| 4°C | −2.88 ± 0.10[bc,A] | −2.70 ± 0.01[c,B] | −2.76 ± 0.20[c,B] | −3.02 ± 0.16[a,A] | −3.16 ± 0.13[a,A] |
| 25°C, RH: 43% | −2.66 ± 0.08[a,B] | −2.70 ± 0.10[a,B] | −2.72 ± 0.17[a,B] | −2.63 ± 0.14[a,B] | −2.72 ± 0.16[a,B] |
| 25°C, RH: 54% | −2.14 ± 0.07[b,C] | −2.42 ± 0.13[ab,C] | −2.30 ± 0.16[ab,C] | −2.31 ± 0.13[ab,C] | −2.63 ± 0.33[a,B] |
| 25°C, RH: 65% | −1.50 ± 0.18[c,D] | −1.66 ± 0.15[bc,D] | −1.81 ± 0.09[ab,D] | −1.69 ± 0.15[bc,D] | −2.05 ± 0.14[a,C] |

**Table 3.** Changes in the $b^*$ value of the bilayer films during storage. Different lower-case and upper-case letters in the table indicate significant differences ($p < 0.05$) in the same row and column, respectively.

| storage conditions | 30 days | 60 days | 90 days | 120 days | 150 days |
|---|---|---|---|---|---|
| −17°C | 17.37 ± 0.42[b,D] | 17.07 ± 0.53[b,D] | 16.87 ± 0.67[b,D] | 16.66 ± 0.26[b,C] | 15.50 ± 0.37[a,C] |
| 4°C | 16.57 ± 0.58[b,CD] | 16.59 ± 0.84[b,D] | 15.53 ± 0.40[ab,C] | 14.65 ± 0.48[a,B] | 14.87 ± 0.71[a,C] |
| 25°C, RH: 43% | 16.01 ± 0.52[a,C] | 14.91 ± 0.72[a,C] | 14.98 ± 0.61[a,B] | 14.95 ± 0.78[a,B] | 14.66 ± 0.87[a,BC] |
| 25°C, RH: 54% | 14.10 ± 0.82[a,B] | 13.49 ± 0.33[a,B] | 14.06 ± 0.79[a,B] | 13.32 ± 0.84[a,A] | 13.32 ± 0.78[a,AB] |
| 25°C, RH: 65% | 12.01 ± 0.55[a,A] | 12.13 ± 0.54[a,A] | 11.57 ± 0.87[a,A] | 12.47 ± 0.45[a,A] | 11.96 ± 1.01[a,A] |

**Table 4.** Changes in the WI value of the bilayer films during storage. Different lower-case and upper-case letters in the table indicate significant differences ($p < 0.05$) in the same row and column, respectively.

| storage conditions | 30 days | 60 days | 90 days | 120 days | 150 days |
|---|---|---|---|---|---|
| −17°C | 81.54 ± 0.49[a,A] | 81.69 ± 0.57[a,A] | 81.74 ± 0.73[a,A] | 82.05 ± 0.21[a,A] | 83.18 ± 0.38[a,A] |
| 4°C | 82.34 ± 0.55[a,AB] | 82.32 ± 0.72[a,A] | 83.35 ± 0.45[ab,B] | 84.04 ± 0.49[b,B] | 83.43 ± 0.68[ab,A] |
| 25°C, RH: 43% | 82.92 ± 0.44[a,B] | 83.87 ± 0.65[a,B] | 83.71 ± 0.50[a,BC] | 83.80 ± 0.76[a,B] | 83.99 ± 0.92[a,AB] |
| 25°C, RH: 54% | 84.80 ± 0.85[a,C] | 85.17 ± 0.24[a,C] | 84.66 ± 0.65[a,C] | 85.35 ± 0.71[a,C] | 85.04 ± 0.62[a,B] |
| 25°C, RH: 65% | 86.68 ± 0.54[a,D] | 86.51 ± 0.55[a,D] | 86.96 ± 0.76[a,D] | 86.17 ± 0.38[a,C] | 86.50 ± 1.02[a,C] |

The WI values of the bilayer films were consistent with the $L^*$, $a^*$ and $b^*$ values, indicating that different storage conditions affected the surface colour of the bilayer films to some extent. However, with prolonged storage time, the surface colour of the bilayer films did not change significantly within 150 days under the same storage conditions.

## 3.2. Effect of storage conditions on the transmittance and haze of the bilayer films

Figure 1 shows the variation in the transmittance of the bilayer films with time under different storage conditions. The transmittance of the bilayer films did not change significantly with storage time,

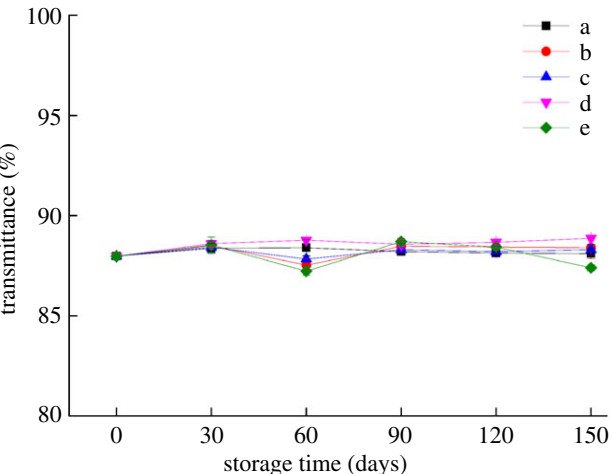

**Figure 1.** Changes in the transmittance of the bilayer films during storage (a–e represent the conditions of −17°C, 4°C and 25°C with 43% RH, 25°C with 54% RH and 25°C with 65% RH, respectively).

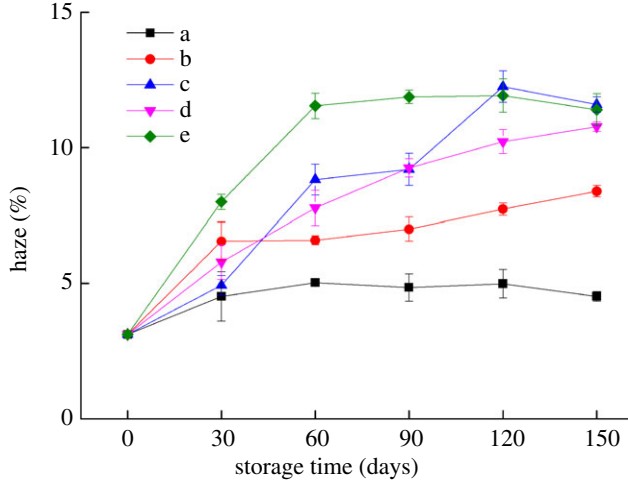

**Figure 2.** Changes in the haze of the bilayer films during storage (a–e represent the conditions of −17°C, 4°C and 25°C with 43% RH, 25°C with 54% RH and 25°C with 65% RH, respectively).

which verifies the conclusions of §3.1. However, when the storage time was 60 or 150 days, the transmittance of the bilayer films fluctuated to varying degrees under refrigeration (4°C) and high RH (65%) conditions. Transmittance usually reflects the change in the surface colour of the film material to some extent and indicates compatibility between the biomacromolecules within the bilayer films. In addition, at 25°C and 54% RH, the transmittance of the bilayer films did not change significantly with the prolongation of storage time [22].

Haze mainly reflects the gloss and transparency of the film material, but haze and transmittance are two different concepts. For example, proper haze helps to increase the consumer acceptability of the film packaging material. Figure 2 shows that the haze of the bilayer films was greater than 0 (variation range: 3.12–11.92%) during different storage periods, because the bilayer films have a certain degree of crystallinity. With prolonged storage time, the haze of the film material increased by varying degrees, which indicates that the biocompatibility of the bilayer films changed during storage. Additionally, the plasticizer glycerin and polyethylene glycol 400 possibly migrated to the surface of the bilayer films during storage, thus increasing both the roughness of the surface of the bilayer films and the haze.

## 3.3. Effect of storage conditions on the moisture content of the bilayer films

For biomacromolecule-based packaging materials, especially polysaccharide edible films, moisture is a good plasticizer. The change in the moisture content of films indirectly affected their barrier and

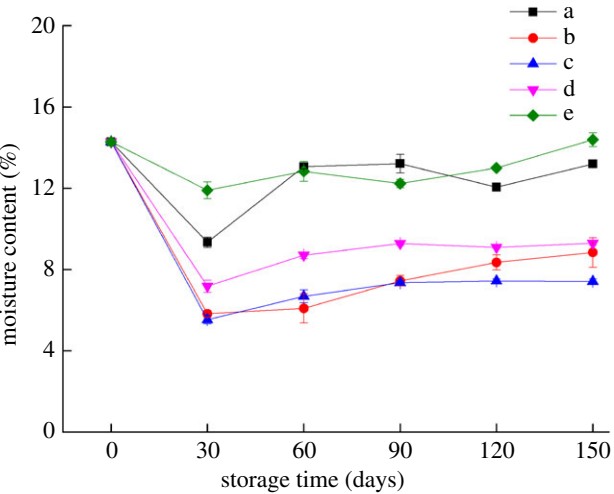

**Figure 3.** Changes in the moisture content of the bilayer films during storage (a–e represent the conditions of −17℃, 4℃ and 25℃ with 43% RH, 25℃ with 54% RH and 25℃ with 65% RH, respectively).

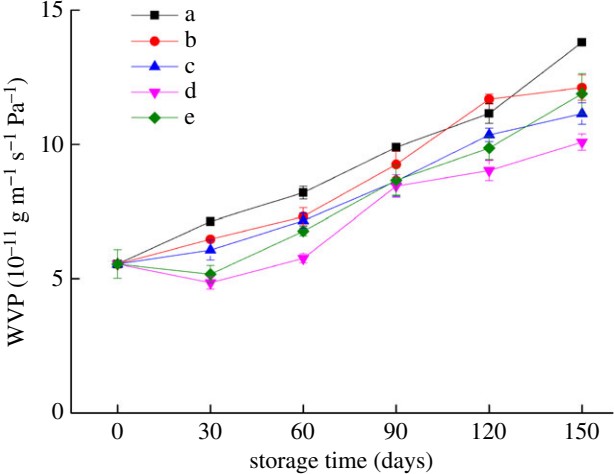

**Figure 4.** Changes in the WVP of the bilayer films during storage (a–e represent the conditions of −17℃, 4℃ and 25℃ with 43% RH, 25℃ with 54% RH and 25℃ with 65% RH, respectively).

mechanical properties. As shown in figure 3, within 0–30 days, the moisture content of the bilayer films decreased significantly ($p < 0.05$), which may be due to the migration of polyethylene glycol 400 during storage, resulting in a decrease in the water holding capacity of the bilayer films. Under the storage conditions of 4℃ and 25℃ with 43% RH, the moisture content of the bilayer films was relatively low, indicating that the moisture content of the edible biomacromolecule films was significantly affected by the external environment, which is a major obstacle to the popularization and application of edible biomacromolecule films. However, with the prolongation of storage time, the moisture content showed an overall increasing trend, reaching 13.19% (−17℃), 8.84% (4℃), 7.41% (25℃ with 43% RH), 9.29% (25℃ with 54% RH) and 14.39% (25℃ with 65% RH) after 150 days. This trend may be due to the oxidation of the Z film and the rejuvenation of the C–W film during storage, resulting in increased cross-linking between the water molecules and crystalline structures in the films [23].

## 3.4. Effect of storage conditions on the WVP of the bilayer films

For starch-based films with strong hydrophilicity, changing the RH of the storage environment greatly influences the properties of the films, especially their water resistance. Figure 4 shows the changes in the WVP with storage time under different storage conditions. After 150 days of storage, the water resistance of the bilayer films became worse. Under the condition of high RH (65%), the WVP of the bilayer film reached $11.88 \times 10^{-11}$ g (m·s·Pa)$^{-1}$ mainly because water played a plasticizing role in the

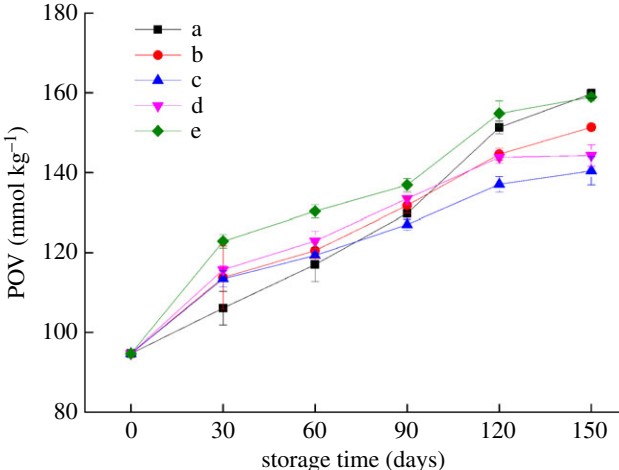

**Figure 5.** Changes in the POV of soya bean oil wrapped in bilayer films during storage (a–e represent the conditions of −17℃, 4℃ and 25℃ with 43% RH, 25℃ with 54% RH and 25℃ with 65% RH, respectively).

bilayer film. Excessive water caused the internal structure of the film to loosen, which allowed the water molecules to penetrate more easily through the film [13]. Under the storage conditions of −17°C and 4°C, the WVP of the bilayer films reached $13.80 \times 10^{-11}$ and $12.11 \times 10^{-11}$ g (m·s·Pa)$^{-1}$ after 150 days of storage, respectively, and was higher than that of the bilayer films stored at 25°C. This was mainly because with increasing temperature, the bonds between polymers became closer and the pore size decreased, which hindered the transfer of water molecules. After 150 days of storage at 25°C and 54% RH, the WVP of the bilayer film was $10.08 \times 10^{-11}$ g (m·s·Pa)$^{-1}$, indicating that its water resistance was relatively good.

## 3.5. Effect of storage conditions on the oxygen barrier properties of the bilayer films

Figure 5 shows that the POV of soya bean oil wrapped in the bilayer films increased significantly after ageing for 150 days ($p < 0.05$), which indicated that the oxygen barrier properties of the bilayer films exhibited a trend of deterioration under different storage conditions. At 25°C and 43% RH, the $POV_{150\,days}$ was lower, reaching 140.47 mmol kg$^{-1}$, while at 25°C and 65% RH, the $POV_{150\,days}$ was higher, reaching 158.97 mmol kg$^{-1}$. Under high RH storage conditions, the oxygen barrier properties of the bilayer films became worse, while under low RH storage conditions, the oxygen barrier of the bilayer films was relatively better. This occurred because in a low RH environment, due to the large difference in RH between the bilayer films and the external environment, the bilayer films lost moisture, causing the internal structure to be more compact and improving the oxygen barrier performance. However, when passing through the bilayer films, water molecules not only bind readily to starch and proteins but also easily establish a strong connection between molecules, allowing the rate of oxygen permeation through the bilayer films to increase dramatically. Under high RH (25°C and 65% RH) storage conditions, the increase in RH resulted in an increase in the moisture content, which may have caused the hydrogen bonds between adjacent molecules to be broken, and the molecular mobility increased, thus leading to the deterioration of the oxygen resistance.

Usually, the oxygen permeability of the films increases with increasing temperature. Under the conditions of this experiment, the oxygen permeability of the bilayer films in the early storage period conformed to this change rule; however, in the late storage period, the results contradicted this expectation. After 150 days, the $POV_{150\,days}$ increased drastically to 159.78 mmol kg$^{-1}$ under the −17°C storage condition, possibly because the passage of oxygen molecules through the bilayer films can be divided into dissolution and diffusion, which are influenced by the changes in the moisture content of the bilayer films. The C–W/Z bilayer films are mainly used in oil packages (electronic supplementary material, The application of the C–W/Z bilayer films) in convenient food industry because of its oxygen barrier.

## 3.6. Effect of storage conditions on the mechanical properties of the bilayer films

The shelf life of packaging materials is directly determined by changes in their mechanical properties during storage. Figures 6 and 7 show that after 30 days of storage at 25°C with RHs of 43, 54, and 65%, the tensile strength of the bilayer films increased to 10.22, 11.04 and 11.07 MPa, respectively,

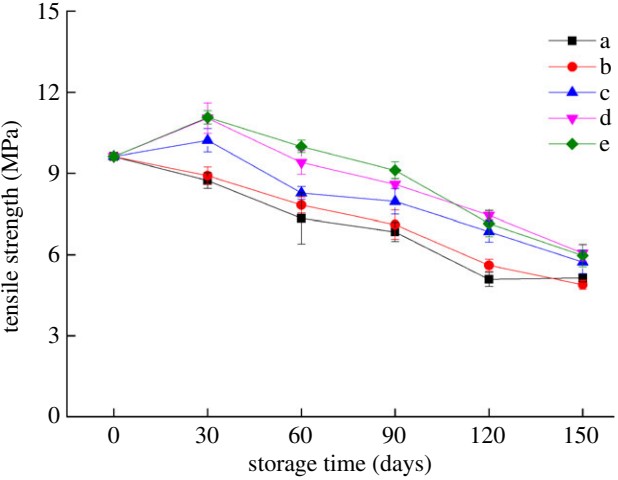

**Figure 6.** Changes in the tensile strength of the bilayer films during storage (a–e represent the conditions of −17℃, 4℃ and 25℃ with 43% RH, 25℃ with 54% RH and 25℃ with 65% RH, respectively).

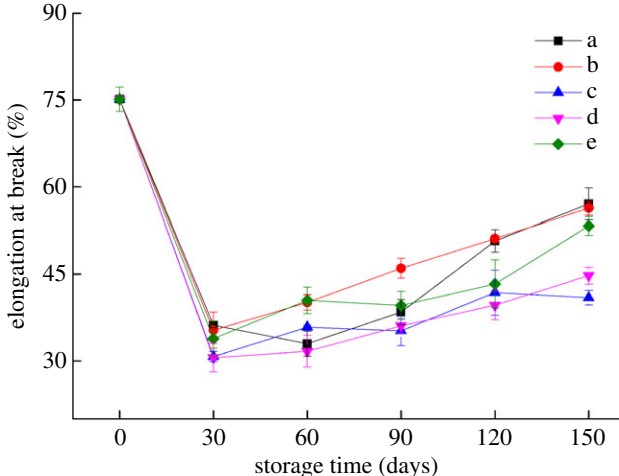

**Figure 7.** Changes in the elongation at break of bilayer films during storage (a–e represent the conditions of −17℃, 4℃ and 25℃ with 43% RH, 25℃ with 54% RH and 25℃ with 65% RH, respectively).

while the elongation at break decreased significantly ($p < 0.05$), which may be because new cross-links still occurred between the C–W and Z molecules during the early storage period [24]. During the 30–150 days storage period, the tensile strength of the bilayer films decreased significantly ($p < 0.05$), especially for bilayer films stored at low temperatures. The reason for this finding may be that at −17℃, the structure of the bilayer films was destroyed due to the freezing, formation and diffusion of ice crystals, leading to a decrease in tensile strength [25]. By contrast, the elongation at break of the bilayer films increased significantly ($p < 0.05$) with prolonged storage time. The change in the moisture content of the bilayer films was also an important factor affecting their mechanical properties because water plays a plasticizing role in bilayer films. With increasing moisture content, water can become embedded between the starch and protein molecules, thus greatly weakening the interaction between or within them, and the structure of the bilayer films becomes more relaxed, resulting in their increased flexibility [26].

## 3.7. Effect of storage conditions on the microstructure of the bilayer films during storage

The changes in the barrier and mechanical properties of the bilayer films reflect the changes in their internal microstructures to a certain extent. Therefore, the changes in the physical and chemical properties of the bilayer films during storage can be explored by SEM. Figures 8–10 show SEM observations of the bilayer film cross-sections under different storage conditions at 0, 30 and

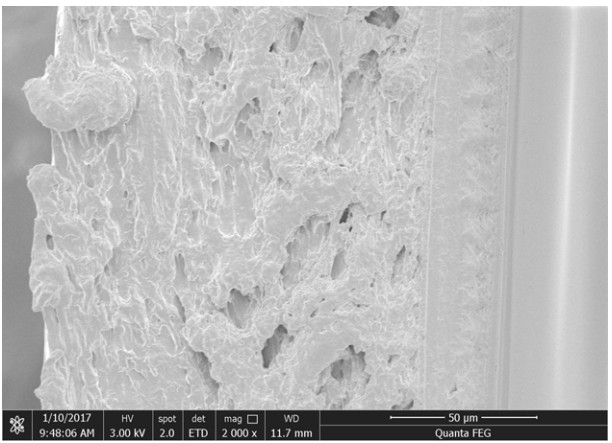

**Figure 8.** SEM image of the cross-section of a bilayer film at 0 days.

(*a*)

(*b*)

(*c*)

(*d*)

(*e*)

**Figure 9.** SEM images of the cross-section of a bilayer film stored for 30 days (*a–e* represent the conditions of −17℃, 4℃, and 25℃ with 43% RH, 25℃ with 54% RH, and 25℃ with 65% RH, respectively).

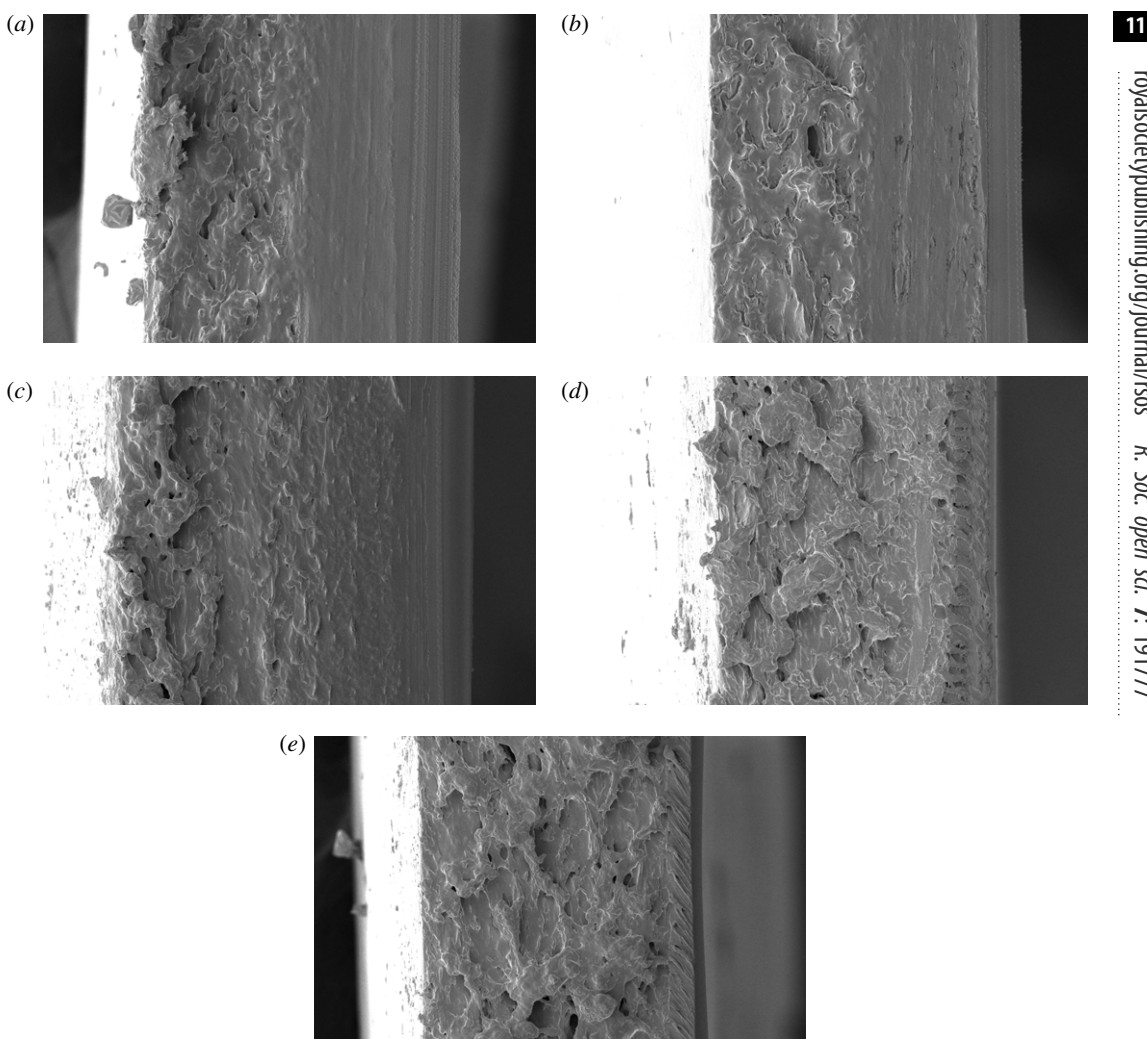

**Figure 10.** SEM images of the cross-section of a bilayer film stored for 150 days (*a*–*e* represent the conditions of −17℃, 4℃, and 25℃ with 43% RH, 25℃ with 54% RH, and 25℃ with 65% RH, respectively).

150 days, respectively. During the storage period of 0–30 days, under −17°C and 25°C with RHs of 43, 54 and 65%, the internal network structure of the bilayer films disappeared, and the films became denser, so that water molecules did not easily penetrate the bilayer films, resulting in a decrease in the moisture content of the films (figures 8 and 9) [27]. In addition, the denser network structure and the change in moisture content may explain the increase in tensile strength and the decrease in elongation at break. Some network structures were still visible in the cross-section of the bilayer film stored at 4°C, but the transition within the bilayer film was natural. After 150 days of storage, some cracks appeared in the bilayer films (figure 10), which allowed water and oxygen molecules to permeate through the bilayer films more easily. This phenomenon explains why the water and oxygen barrier performance of the bilayer film became worse in the later period of storage [28]. In addition, the changing trends in the cross-sectional microstructure of the bilayer films verified the changes in the moisture content and mechanical properties of the films [29].

## 4. Conclusion

The WI value of the bilayer films increased with increasing temperature and RH during the same storage period, but there was no significant change in the colour of the bilayer films under the same storage conditions. After 60 days of storage, the transmittance and haze of the bilayer films changed greatly, and the haze of the bilayer films stored at 25°C was more obvious than that stored at low temperature. With prolonged storage time, the water resistance and oxygen barrier properties of the

bilayer films showed a trend of deterioration. Under storage conditions of 25°C and 54% RH, the water resistance performance of the bilayer film was relatively good, while the bilayer film stored at 25°C and 43% RH had better oxygen barrier properties. During storage, the bilayer film stored at 25°C with 65% RH had relatively good tensile strength, while the bilayer film stored at low temperature had relatively good elongation at break. The SEM results showed that the bonding between bilayer films was tight and that the internal network structure of the bilayer films reappeared after disappearing. The changing trend of the internal microstructure verified the changing trends of the barrier performance and mechanical properties of the bilayer films. These results contribute to the application of the edible bilayer film, a kind of new packaging material, in the food industry.

Data accessibility. Data available from the Dryad Digital Repository: https://dx.doi.org/10.5061/dryad.sqv9s4n07 [30].

Competing interests. We have no competing interests.

Funding. This work was supported by National Natural Science Foundation of China (grant no. 21676073), and the 13th Five-year National Key Research and Development Plan (grant no. 2018YFD0401100).

Acknowledgements. We thank Yu Liu for the assistance with SEM tests.

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
