## [Reviewer comments · Royal Society Open Science]

Review History

RSOS-191777.R0 (Original submission)

Review form: Reviewer 1

Is the manuscript scientifically sound in its present form?

Yes

Are the interpretations and conclusions justified by the results?

Yes

Is the language acceptable?

Yes

Do you have any ethical concerns with this paper?

No

Have you any concerns about statistical analyses in this paper?

No

Recommendation?

Accept with minor revision (please list in comments)

Comments to the Author(s)

This manuscript describes the effects of different storage temperature, relative humidity and time on the surface colour, transmittance, haze, moisture content, water vapour permeability (WVP), oxygen barrier properties, tensile strength and elongation at break of corn starch-wheat starch/zein edible bilayer films stored under different storage conditions. Although it reports some important results, revision is necessary.

My opinions and comments about this manuscript are as follows:

1) Line 53: After this sentence, I suggest the authors that the following sentence and references should be given.

The additive ingredients added into the film matrix affect the chemical and physical properties of film packaging materials [11,12]. In addition, physical and chemical changes may occur ...".

- [11] LWT-Food Science and Technology, 44 (2011) 465-472.
- [12] Carbohydrate Polymers, 150 (2016) 259-268.

2) Line 54: Please check the spelling of "humiditie".

3) Lines 58-59: At the end of this sentence, please give these studies as references.

4) Line 120: Abbreviation "WVP" should be defined at first mention.

5) Lines 129 and 130: Please check the meaning of symbol "m" used in Eq. (2).

Review form: Reviewer 2

Is the manuscript scientifically sound in its present form?

Yes

Are the interpretations and conclusions justified by the results?

Yes

Is the language acceptable?

Yes

Do you have any ethical concerns with this paper?

No

Have you any concerns about statistical analyses in this paper?

No

Recommendation?

Major revision is needed (please make suggestions in comments)

Comments to the Author(s)

The manuscript entitled: "Effect of storage condition on the physicochemical properties of corn-wheat starch / zein edible bilayer films" is a work dealing with the preparation of packaging materials and their properties after long-time investigation to prove their stability. The authors have designed a series of experiments dealing with the optical, mechanical as well as transport properties of the materials prepared.

The results are well presented and well performed while also, the data provided are not indicating significant errors or uncertainties.

Nevertheless, the manuscript is not presenting the properties of the packaging materials and their function as membrane - a barrier for different substances (e.g., gases from the packaging production or bacteria). I believe that the manuscript can provide some information or statement also for this point.

Additional points that need attention are the points of aging testing. Are all the different materials aged under the same conditions? Was it possible to expose the materials, e.g., under controlled irradiation to enhance the aging effect and to see what is happening in cases of use of the packaging materials at different environments (e.g., different countries?).

What is the correlation of the optical properties with the mechanical properties?

What is the correlation between the oxygen transport properties with the WI measurements?

What about the N₂ permeability of CO₂?

Final and not least, the structure of the prepared films is losing significantly the porosity and foamy structure. Did the authors check on the compression effects of the films? Was it possible to perform nonindentation measurements to see the difference before and after aging - storage?

Decision letter (RSOS-191777.R0)

10-Dec-2019

Dear Mr Chen:

Title: Effect of storage condition on the physicochemical properties of corn-wheat starch / zein edible bilayer films

Manuscript ID: RSOS-191777

The editor assigned to your manuscript has now received comments from reviewers. We would like you to revise your paper in accordance with the referee and Subject Editor suggestions which can be found below (not including confidential reports to the Editor). Please note this decision does not guarantee eventual acceptance.

Please submit your revised paper before 02-Jan-2020. Please note that the revision deadline will expire at 00.00am on this date. If we do not hear from you within this time then it will be assumed that the paper has been withdrawn. In exceptional circumstances, extensions may be possible if agreed with the Editorial Office in advance. We do not allow multiple rounds of revision so we urge you to make every effort to fully address all of the comments at this stage. If deemed necessary by the Editors, your manuscript will be sent back to one or more of the original reviewers for assessment. If the original reviewers are not available we may invite new reviewers.

When submitting your revised manuscript, you must respond to the comments made by the referees and upload a file "Response to Referees" in "Section 6 - File Upload". Please use this to document how you have responded to the comments, and the adjustments you have made. In

order to expedite the processing of the revised manuscript, please be as specific as possible in your response.

RSC Associate Editor:
Comments to the Author:
(There are no comments.)

RSC Subject Editor:
Comments to the Author:
(There are no comments.)

Reviewers' Comments to Author:
Reviewer: 1

Comments to the Author(s)

This manuscript describes the effects of different storage temperature, relative humidity and time on the surface colour, transmittance, haze, moisture content, water vapour permeability (WVP), oxygen barrier properties, tensile strength and elongation at break of corn starch-wheat starch/zein edible bilayer films stored under different storage conditions. Although it reports some important results, revision is necessary.

My opinions and comments about this manuscript are as follows:

1) Line 53: After this sentence, I suggest the authors that the following sentence and references should be given.

The additive ingredients added into the film matrix affect the chemical and physical properties of film packaging materials [11,12]. In addition, physical and chemical changes may occur ...".

- [11] LWT-Food Science and Technology, 44 (2011) 465-472.
- [12] Carbohydrate Polymers, 150 (2016) 259-268.

2) Line 54: Please check the spelling of "humiditie".

3) Lines 58-59: At the end of this sentence, please give these studies as references.

4) Line 120: Abbreviation "WVP" should be defined at first mention.

5) Lines 129 and 130: Please check the meaning of symbol "m" used in Eq. (2).

Reviewer: 2

Comments to the Author(s)

The manuscript entitled: "Effect of storage condition on the physicochemical properties of corn-wheat starch / zein edible bilayer films" is a work dealing with the preparation of packaging materials and their properties after long-time investigation to prove their stability. The authors have designed a series of experiments dealing with the optical, mechanical as well as transport properties of the materials prepared.

The results are well presented and well performed while also, the data provided are not indicating significant errors or uncertainties.

Nevertheless, the manuscript is not presenting the properties of the packaging materials and their function as membrane - a barrier for different substances (e.g., gases from the packaging production or bacteria). I believe that the manuscript can provide some information or statement also for this point.

Additional points that need attention are the points of aging testing. Are all the different materials aged under the same conditions? Was it possible to expose the materials, e.g., under controlled irradiation to enhance the aging effect and to see what is happening in cases of use of the packaging materials at different environments (e.g., different countries?).

What is the correlation of the optical properties with the mechanical properties?

What is the correlation between the oxygen transport properties with the WI measurements?

What about the N₂ permeability of CO₂?

Final and not least, the structure of the prepared films is losing significantly the porosity and foamy structure. Did the authors check on the compression effects of the films? Was it possible to perform nonindentation measurements to see the difference before and after aging - storage?

Author's Response to Decision Letter for (RSOS-191777.R0)

See Appendix A.

RSOS-191777.R1 (Revision)

Review form: Reviewer 2

Is the manuscript scientifically sound in its present form?

Yes

Are the interpretations and conclusions justified by the results?

Yes

Is the language acceptable?

Yes

Do you have any ethical concerns with this paper?

No

Have you any concerns about statistical analyses in this paper?

Yes

Recommendation?

Accept as is

Comments to the Author(s)

The revised manuscript is now OK for publication.

Decision letter (RSOS-191777.R1)

13-Jan-2020

Dear Mr Chen:

Title: Effect of storage condition on the physicochemical properties of corn-wheat starch / zein edible bilayer films

Manuscript ID: RSOS-191777.R1

It is a pleasure to accept your manuscript in its current form for publication in Royal Society Open Science. The chemistry content of Royal Society Open Science is published in collaboration with the Royal Society of Chemistry.

RSC Associate Editor:
Comments to the Author:
(There are no comments.)

RSC Subject Editor:
Comments to the Author:
(There are no comments.)

Reviewer(s)' Comments to Author:
Reviewer: 2

Comments to the Author(s)
The revised manuscript is now OK for publication.

Appendix A

Manuscript Draft

Manuscript ID: RSOS-191777

Title: Effect of storage condition on the physicochemical properties of corn-wheat starch / zein edible bilayer films

Article Type: Research

Keywords: Corn-wheat starch;
Zein;
Bilayer films;
Storage conditions;
Physicochemical properties

Abstract: The functional properties of biopolymer-based film packaging materials are susceptible to external storage conditions. The effects of different storage temperature, relative humidity (RH) and duration on the apparent form, barrier properties, mechanical properties and microstructure of corn-wheat starch/zein bilayer films were studied. From 0 to 150 d, storage temperature and RH, but not storage time, affected the appearance and colour of the bilayer films. The increase in haze of the bilayer films stored at 25°C was much greater than that at low temperatures. With increased storage time, the moisture content first increased and then decreased, while the water resistance and oxygen barrier properties of the bilayer films worsened. After 150 d, the bilayer film stored at 25°C with 54% RH had better water resistance properties. The oxygen barrier properties of the bilayer film stored at 25°C with 43% RH were preferable to those of other groups because the peroxide value of vegetable oil packed in the former bilayer film was the lowest. The tensile strength of bilayer films stored at 25°C with RH of 43%, 54%, and 65% decreased, but were still better than those stored at low temperatures (-17°C, 4°C), which were tough due to their high elongation at break. SEM results showed tight bonds between the bilayer films, and the network structure inside the films disappeared and reappeared during storage. The cross-sectional compactness changed, and there was no film separation after 150 d.

<Journal Name>: Royal Society Open Science

< Manuscript ID>: RSOS-191777

<Manuscript Title>: Effect of storage condition on the physicochemical properties of corn-wheat starch / zein edible bilayer films

Dear Editor Dr. Laura Smith:

Thank you very much for your kind information. We highly appreciate the meaningful comments from editor and reviewers concerning our manuscript entitled “Effect of storage condition on the physicochemical properties of corn-wheat starch / zein edible bilayer films”. Those comments are all valuable and very helpful for revising and improving our paper, as well as the important guiding significance to our researches. We have carefully considered the comments and have tried our best to revise the manuscript according to the comments. The amendments are **highlighted in red** in the revised manuscript. Enclosed please find the revised version, which we would like to submit for your kind consideration. Also, a point-by-point list of our response to the reviewers’ and Editor’s comments have been attached.

We would like to express our great appreciation to you and reviewers for comments on our paper. Looking forward to hearing from you.

Thank you and best regards,

Sincerely yours,

Fusheng Chen, Ph.D.

Professor

Henan University of Technology

Zhengzhou, 450001, P.R. China

E-mail: fushengc@haut.edu.cn

Incl.: Responses (in blue) to the comments from reviewers.

Response to Referees

Reviewer #1:

Comments to the Author(s)

This manuscript describes the effects of different storage temperature, relative humidity and time on the surface colour, transmittance, haze, moisture content, water vapour permeability (WVP), oxygen barrier properties, tensile strength and elongation at break of corn starch-wheat starch/zein edible bilayer films stored under different storage conditions. Although it reports some important results, revision is necessary.

My opinions and comments about this manuscript are as follows:

Question 1: Line 53: After this sentence, I suggest the authors that the following sentence and references should be given.

The additive ingredients added into the film matrix affect the chemical and physical properties of film packaging materials [11,12]. In addition, physical and chemical changes may occur ...”.

- [11] LWT-Food Science and Technology, 44 (2011) 465-472.
- [12] Carbohydrate Polymers, 150 (2016) 259-268.

Authors' response: Thanks for reviewer's useful comment. We have added the sentence and references recommended by the reviewer. Please refer to line 53-55 and line 343-348 in the revised edition.

Question 2: Line 54: Please check the spelling of “humiditie”.

Authors' response: Thanks for reviewer's careful comment. We are sorry about the spelling mistake. We have replaced “humiditie” with “humidity”. In addition, we have replaced “durations”, “concentrations”, “intensities” with “duration”, “concentration”, “intensity”, respectively. Please refer to line 56 in the revised edition.

Question 3: Lines 58-59: At the end of this sentence, please give these studies as references.

Authors' response: Thanks for reviewer's careful comment. We have added two references after the sentence “there are few studies on the changes in the functional properties of bilayer films during storage”. Please refer to reference [17, 18] in line 60 and line 361-367 in the revised edition.

Question 4: Line 120: Abbreviation “WVP” should be defined at first mention.

Authors' response: Thanks for reviewer's kind suggestion. The abbreviation “WVP” has been defined at first mention. Please refer to line 65 in the revised edition.

Question 5: Lines 129 and 130: Please check the meaning of symbol “m” used in Eq. (2).

Authors' response: Thanks for reviewer's insightful comment. We apologize for this expression error. We have revised this sentence to “*m* represents the increased mass of anhydrous calcium chloride after water absorption”. Additionally, in this section, we have removed the extra word “film” in line 124. Please refer to line 124 and line 130-131 in the revised edition.

Reviewer #2:

Comments to the Author(s)

The manuscript entitled: “Effect of storage condition on the physicochemical properties of corn-wheat starch / zein edible bilayer films” is a work dealing with the preparation of packaging materials and their properties after long-time investigation to prove their stability. The authors have designed a series of experiments dealing with the optical, mechanical as well as transport properties of the materials prepared.

The results are well presented and well performed while also, the data provided are not indicating significant errors or uncertainties.

Question 1: Nevertheless, the manuscript is not presenting the properties of the packaging materials and their function as membrane-a barrier for different substances (e.g., gases from the packaging production or bacteria). I believe that the manuscript can provide some information or statement also for this point.

Authors’ response: Thanks for reviewer’s insightful comment. To present the properties of the packaging materials and their function as films, the sentence “The C-W/Z bilayer films are mainly used in oil packages (Electronic Supplementary Material, The application of the C-W/Z bilayer films) in convenient food industry because of its oxygen barrier.” was added after “...which are influenced by the changes in the moisture content of the bilayer films”. Please refer to line 248-250 in the revised edition.

In addition, the application of the C-W/Z bilayer films as oil packages was studied. This section was provided as Electronic Supplementary Material “The application of the C-W/Z bilayer films”. Please refer to the Electronic Supplementary Material for more details.

Question 2: Additional points that need attention are the points of aging testing. Are all the different materials aged under the same conditions? Was it possible to expose the materials, e.g., under controlled irradiation to enhance the aging effect and to see what is happening in cases of use of the packaging materials at different environments (e.g., different countries?).

Authors’ response: Thanks for reviewer’s careful comment. During the use of the materials, due to the combined effects of environmental factors such as heat, oxygen, water, light, microorganisms, and chemical media, a series of changes will occur in the structure of the materials. And the physical and chemical properties of the materials will also deteriorate accordingly, such as hardening, sticking, embrittlement, discoloration, loss of strength and so on. These changes and phenomena are called aging. The essence of material aging is the change of its physical or chemical structure. Aging of materials is an irreversible change, which is a common problem of materials. In our study, all the corn-wheat starch / zein edible bilayer films were prepared using same methods before the storage. Then, all the same bilayer films were aged under different storage conditions respectively. The variables (storage temperature, relative humidity and storage time) were chosen based on the possible environment in which the C-W/Z bilayer film oil packages may be located. Meanwhile, previous studies showed that these variables affected the performance of the films as well [1-3].

Furthermore, it is possible to expose the materials or control the amount of light irradiation to enhance the aging effect. Previous studies showed that light can cause material aging [4, 5]. This comment also provides an important reference for our future research to investigate the effects of light type, intensity, and time on the bilayer film performance.

Due to geographical location and climate, the environment of different countries is quite different, such as temperature, humidity, air pressure, illumination time and so on. Even in the same country, the environment in different regions will be quite different. For example, some countries near the equator have high temperature, heavy rainfall and high humidity, while some countries in Africa have high temperature, low rainfall and low humidity. The structure and function of the film materials may change differently in different environments (e.g., different countries). Therefore, it is meaningful to study the changes in the properties of the film materials in different environments, which is the main purpose of our study. The environmental variables in this research (storage temperature, relative humidity and storage time) were chosen based on the possible environment in which the C-W/Z bilayer film oil package may be located. Moreover, in our following work, we would simulate the typical environmental conditions in different countries and test the structural and functional changes of the bilayer films under these environmental conditions.

Question 3: What is the correlation of the optical properties with the mechanical properties?

Authors' response: Thanks for reviewer's insightful comment. There may be some correlations between optical properties and mechanical properties. The optical properties reflect the surface morphology and internal network structure of the films, which affect the mechanical properties of the films to some extent. The mechanical properties of films represent their ability to maintain their integrity and endure external stress during the processing, transportation, handling and storage of packaged materials. Sufficient mechanical strength and extensibility are generally needed for use in food packaging applications [6, 7].

Generally, the films with high transmittance may have low surface roughness, smooth internal structure and high tensile strength, while the films with low transmittance may have high surface roughness, rough internal structure and poor tensile strength. Similar results were found in Pérez's study [8]. In addition, Artharn et al. [3] found that the change trend of transmittance was in agreement with the tensile strength in mechanical properties of the films, which may be due to the increased crosslinking of the film matrix. The specific relationship between optical and mechanical properties needs further study.

Question 4: What is the correlation between the oxygen transport properties with the WI measurements?

Authors' response: Thanks for reviewer's useful comment. There may be some correlations between oxygen transport properties and WI measurements. Colour is an attribute of fundamental importance for films and is greatly affected by several factors including plasticizer addition, thermal treatment, fabrication process, and storage conditions [8]. Oxygen permeability is the next most commonly studied transport property of edible films after water vapor permeability [9]. The oxygen permeability of films is influenced by various factors, such as relative humidity, plasticizer, and polymer morphology [10].

Similar to Question 3, the correlation between oxygen transport properties and WI measurements may be caused by the internal structure of the films. As can be seen from the results of this study (Table 4 and Fig. 5), with the increase of the relative humidity in the same storage time, the variation trend of the WI values of the three films stored at 25°C was basically consistent with the oxygen transport properties. However, in the low temperature groups (-17°C, 4°C), there was no obvious correlation between the WI and oxygen transport properties. This may be because the

oxygen permeability is not only related to the WI values, but also to the temperature.

Question 5: What about the N₂ permeability of CO₂?

Authors' response: Thanks for reviewer's useful comment. Oxygen permeability is one of the most commonly studied transport property of edible films. Limited oxygen diffusion and migration within the films is desirable because the presence of oxygen is usually associated with oxidation of food, which leads to flavor changes, odor developments, and nutritional losses [9]. In this study, we observed that the C-W/Z bilayer films exhibited good barrier properties to oxygen. Therefore, the C-W/Z bilayer films were mainly prepared as oil packages to inhibit oil oxidation. So, the oxygen permeability of the films was mainly measured in this study. If necessary, we will combine the actual application environment of the C-W/Z bilayer films in the future research to supplement the measurement of N₂ permeability of CO₂.

Question 6: Final and not least, the structure of the prepared films is losing significantly the porosity and foamy structure. Did the authors check on the compression effects of the films? Was it possible to perform nanoindentation measurements to see the difference before and after aging-storage?

Authors' response: Thanks for reviewer's precious comment. Although in the current experiment, we have not yet checked the compression effect, we will take the compression effect as a measurement indicator and observe the change of the film structure after compression in future studies.

Moreover, it was possible to perform nanoindentation measurements to see the difference before and after aging-storage. In the case that the traditional mechanical test can not meet the measurement requirements, it has become the focus of scholars to seek more advanced mechanical test methods. Nanoindentation is an emerging method of loading, unloading and recording the changes of indentation area and displacement on the material surface with an indenter, and then obtaining the mechanical properties of the materials. Because of its precise displacement and load resolution, nanoindentation is often used to test the mechanical properties of materials at the nanoscale, including elastic modulus, hardness, fatigue characteristics and so on [11, 12]. At present, nanoindentation is mostly applied to the inedible film materials [13-15], while the application in edible film materials are less studied. Escamilla-García [16] et al. measured the elastic modulus and hardness of the zein and chitosan edible films by nanoindentation measurements. Therefore, in the future research, we can try to use nanoindentation measurements to see the difference before and after aging-storage.

References

1. Tian K, Shao Z, Chen X. 2011 Investigation on thermally-induced conformation transition of soy protein film with variable-temperature FTIR spectroscopy. *J. Appl. Polym. Sci.* **124**, 2838-2845. (DOI: 10.1002/app.35309)
2. Osés J, Fernández-Pan I, Mendoza M, Maté JI. 2009 Stability of the mechanical properties of edible films based on whey protein isolate during storage at different relative humidity. *Food Hydrocolloids.* **23**, 125-131. (DOI: 10.1016/j.foodhyd.2007.12.003)
3. Artharn A, Prodpran T, Benjakul S. 2009 Round scad protein-based film: Storage stability and its effectiveness for shelf-life extension of dried fish powder. *LWT-Food Sci. Technol.* **42**, 1238-1244. (DOI: 10.1016/j.lwt.2008.08.009)

4. Deng Z, Wang M, Zhu C, Li C, Liu J, Tu M, Xie L, Gui D. 2019 Study on light aging of anhydride-cured epoxy resin used for RGB LED packaging material. *Polym. Test.* **80**, 106131. (DOI: 10.1016/j.polymertesting.2019.106131)
5. Lowe BJ, Smith CA, Fraser-Miller SJ, Paterson RA, Daroux F, Ngarimu-Cameron R, Ford B, Gordon KC. 2017 Light-ageing characteristics of Māori textiles: Colour, strength and molecular change. *J. Cult. Herit.* **24**, 60-68. (DOI: 10.1016/j.culher.2016.11.012)
6. Nouraddini M, Esmaili M, Mohtarami F. 2018 Development and characterization of edible films based on eggplant flour and corn starch. *Int. J. Biol. Macromol.* **120**, 1639-1645. (DOI: 10.1016/j.ijbiomac.2018.09.126)
7. Zhou X, Yang R, Wang B, Chen K. 2019 Development and characterization of bilayer films based on pea starch/polylactic acid and use in the cherry tomatoes packaging. *Carbohydr. Polym.* **222**, 114912. (DOI: 10.1016/j.carbpol.2019.05.042)
8. Pérez L, Piccirilli G, Delorenzi N, Verdini R. 2016 Effect of different combinations of glycerol and/or trehalose on physical and structural properties of whey protein concentrate-based edible films. *Food Hydrocolloids.* **56**, 352-359. (DOI: 10.1016/j.foodhyd.2015.12.037)
9. Miller KS, Krochta JM. 1997 Oxygen and aroma barrier properties of edible films: A review. *Trends Food Sci. Technol.* **8**, 228-237. (DOI: 10.1016/S0924-2244(97)01051-0)
10. Ferreira ARV, Torres CAV, Freitas F, Sevrin C, Grandfils C, Reis MAM, Alves VD, Coelho IM. 2016 Development and characterization of bilayer films of FucoPol and chitosan. *Carbohydr. Polym.* **147**, 8-15. (DOI: 10.1016/j.carbpol.2016.03.089)
11. Sahay S, Pandey MK, Kar AK. 2019 Metal concentration dependent mechanical properties of electrodeposited nickel incorporated diamond like carbon (Ni-DLC) thin films studied by nanoindentation. *Appl. Surf. Sci.* **489**, 73-79. (DOI: 10.1016/j.apsusc.2019.05.256)
12. Jia H, Wang G, Chen S, Gao Y, Li W, Liaw PK. 2018 Fatigue and fracture behavior of bulk metallic glasses and their composites. *Prog. Mater. Sci.* **98**, 168-248. (DOI: 10.1016/j.pmatsci.2018.07.002)
13. Guo C, Fang Y, Chen F, Feng T. 2019 Nanoindentation creep behavior of electrodeposited Ni-P nanoglass films. *Intermetallics.* **110**, 106480. (DOI: 10.1016/j.intermet.2019.106480)
14. Ramos-Fernández JM, Beleña I, Romero-Sánchez MD, Fuensanta M, Guillem C, López-Buendía ÁM. 2012 Study of the film formation and mechanical properties of the latexes obtained by miniemulsion co-polymerization of butyl acrylate, methyl acrylate and 3-methacryloxypropyltrimethoxysilane. *Prog. Org. Coat.* **75**, 86-91. (DOI: 10.1016/j.porgcoat.2012.03.012)
15. Zheng Q, Li Z, Yang J, Kim JK. 2014 Graphene oxide-based transparent conductive films. *Prog. Mater. Sci.* **64**, 200-247. (DOI: 10.1016/j.pmatsci.2014.03.004)
16. Escamilla-García M, Calderón-Domínguez G, Chanona-Pérez JJ, Farrera-Rebollo RR, Andraca-Adame JA, Arzate-Vázquez I, Mendez-Mendez JV, Moreno-Ruiz LA. 2013 Physical and structural characterisation of zein and chitosan edible films using nanotechnology tools. *Int. J. Biol. Macromol.* **61**, 196-203. (DOI: 10.1016/j.ijbiomac.2013.06.051)